# Mitral surgical redo versus transapical transcatheter mitral valve implantation

**Alina Zubarevich**[1]*, **Marcin Szczechowicz**[1], **Arian Arjomandi Rad**[2],
**Robert Vardanyan**[2], **Philipp Marx**[1], **Alexander Lind**[3], **Rolf Alexander Jánosi**[3],
**Mehdy Roosta-Azad**[1], **Rizwan Malik**[1], **Markus Kamler**[1], **Matthias Thielmann**[1],
**Mohamed El Gabry**[1], **Bastian Schmack**[1], **Arjang Ruhparwar**[1], **Alexander Weymann**[1],
**Daniel Wendt**[1]

**1** Department of Thoracic and Cardiovascular Surgery, West German Heart and Vascular Center, University of Duisburg-Essen, Essen, Germany, **2** Department of Medicine, Faculty of Medicine, Imperial College London, London, United Kingdom, **3** Department of Cardiology & Vascular Medicine, West German Heart and Vascular Center, University of Duisburg-Essen, Essen, Germany

* alina.zubarevich@gmail.com

**Data Availability Statement:** All relevant data are within the paper and its Supporting Information files.

## Abstract

### Background

Transcatheter methods have been rapidly evolving to provide an alternative less invasive therapeutic option, mainly because redo patients often present with multiple comorbidities and high operative risk. We sought to evaluate and compare our experience with transapical transcatheter mitral valve replacement (TA-TMVR) to conventional redo mitral valve replacement in patients presenting with degenerated biological mitral valve prostheses or failed valve annuloplasty.

### Methods and material

Between March 2012 and November 2020, 74 consecutive high-risk patients underwent surgical redo mitral valve replacement (n = 33) or TA-TMVR (n = 41) at our institution. All patients presented with a history of a surgical mitral valve procedure. All transcatheter procedures were performed using the SAPIEN XT/3™ prostheses. Data collection was prospectively according to MVARC criteria.

### Results

The mean logistic EuroSCORE-II of the whole cohort was 19.9±16.7%, and the median STS-score was 11.1±12.5%. The mean age in the SMVR group was 63.7±12.8 years and in the TMVR group 73.6±9.7 years. Patients undergoing TA-TMVR presented with significantly higher risk scores. Echocardiography at follow up showed no obstruction of the left ventricular outflow tract, no paravalvular leakage and excellent transvalvular gradients in both groups (3.9±1.2 mmHg and 4.2±0.8 mmHg in the surgical and transcatheter arm respectively). There was no difference in postoperative major adverse events between the groups with no strokes in the whole cohort. Both methods showed similar survival rates at one year and a 30-day mortality of 15.2% and 9.8% in SAVR and TMVR group, respectively. Despite

**Funding:** We acknowledge support by the Open Access Publication Fund of the University of Duisburg-Essen.

**Competing interests:** Daniel Wendt is working as a proctor for Edwards Lifesciences. Other authors have no conflicts of interest. This does not alter our adherence to PLOS ONE policies on sharing data and materials.

using contrast dye in the transcatheter group, the rate of postoperative acute kidney failure was similar between the groups.

## Conclusion

Despite several contraindications for surgery, we showed the non-inferiority of TA-TMVR compared to conventional surgical redo procedures in high-risk patients. With its excellent hemodynamic and similar survival rate, TA-TMVR offers a feasible alternative to the conventional surgical redo procedure in selected patients.

## Introduction

Redo surgical mitral valve replacement (SMVR) remains the gold standard treatment in patients who previously underwent a surgical mitral valve (MV) procedure and are presenting with recurrent mitral valve pathologies. Current literature indicates that surgical redo procedures might be required in up to 35% of patients who have undergone MV surgery [1]. Over the past decades, transcatheter methods have been rapidly evolving to provide an alternative less invasive therapeutic option, mainly owing to the fact that redo patients often present with multiple comorbidities and a high operative risk. Although, recent studies have demonstrated the feasibility of transcatheter methods in redo valve procedures, there are only few large studies comparing surgical and interventional methods [2]. The aim of this study is to evaluate our experience with surgical redo SMVR and transapical transcatheter mitral valve replacement (TA-TMVR).

## Methods and materials

### Study design and populations

Between March 2012 and November 2020, 74 consecutive patients underwent a surgical redo SMVR or TA-TMVR at our institution. We analyzed and compared the outcomes and postoperative complications in patients undergoing a surgical redo SMVR operation or TA-TMVR using the SAPIEN XT™ or SAPIEN 3™ transcatheter heart valve (Edwards Lifesciences, Irvine, CA, USA). Patients were included, if they required a redo mitral valve procedure, presenting either with dysfunctional biological mitral valve prosthesis or with failed ring-annuloplasty with semirigid continuous annuloplasty rings. Patients were excluded, if the underlying disease was infective endocarditis of the mitral valve, and if concomitant coronary artery bypass CABG procedure was needed.

Our interdisciplinary Heart Team discussed all patients. Postoperative echocardiographic evaluation of the implanted valve prosthesis function was performed at our institution at hospital discharge and during follow-up. Data was collected prospectively as a part of our institutional database, including detailed information on patients' demographics; baseline clinical characteristics; laboratory, echocardiographic, and hemodynamic parameters; intraoperative variables; and postoperative outcomes. The study was conducted according to the Declaration of Helsinki, as revised in 2013. The ethical board of our institution approved the study protocol and data gathering (Ethics committee University Duisburg-Essen, approval number: 21-9937-BO) and waived the patients' individual informed consent. All patients signed the informed consent on follow-up at hospital admission.

## Operative techniques

All transcatheter procedures were performed via transapical access, under general anesthesia in the presence of our institutional Heart Team in a special equipped hybrid operating room. The standard access route for valve-in-valve and valve-in-ring mitral valve procedures at our institution is the direct transapical approach. Therefore, in all of our patients in the TMVR group, transapical approach was used. The transapical access was performed as previously described by our group [3] using 4 pledged-armed U-stiches (Prolene 3–0, MH needle). In brief, access to the left ventricular apex was obtained by a 4–6 cm anterolateral minithoracotomy in the fourth, fifth or sixth intercostal space. Heparin was administered with an intended activated clotting time (ACT) > 250s [3, 4]. After puncturing the apex, a soft guidewire was advanced under fluoroscopic guidance into the right pulmonary vein across the diseased mitral valve. Then, via a Pigtail catheter, an Extra-Stiff wire for further guidance exchanged the soft wire, and a transapical sheath was advanced. The reversely crimped transcatheter valve was finally deployed under ventricular overpacing (120 bpm) [5]. The landing zone was identified mainly with fluoroscopic guidance. Device function was evaluated by transesophageal echocardiography.

For SMVR, the heart was accessed via redo median sternotomy. Cardiopulmonary bypass (CPB) was initiated with the direct cannulation of the ascending aorta and bicaval cannulation of the right atrium. Moderate hypothermic cardiac arrest at 32˚C was performed for all procedures. Myocardial protection was achieved with cold crystalloid cardioplegia. The mitral valve was exposed through left atriotomy via the Waterson's groove or through the right atriotomy and atrial septostomy if any tricuspid valve procedure had to be performed. Extensive debridement of the mitral annulus was performed under preservation of the chords if possible. The MV prosthesis (mechanical or biological) was inserted with single horizontal 4–0 Ethibond pledgeted sutures directed from the left ventricle into the left atrium. After assessment of the valve performance and careful de-airing, the patient was weaned from CPB.

Concomitant aortic valve procedures were performed prior to the MV implantation and tricuspid valve procedures were performed after the MV procedure on the beating heart.

## ViV and ViR sizing

The 'valve-in-valve app', developed by Bapat et al. and the company UBQO was used for design and sizing information of the pre-existing specific mitral bioprosthesis or annuloplasty ring [6]. Multidetector computed tomography (MDCT) was performed on a Somatom™ dual-source force CT (2x192 slices, Siemens Medical Systems, Erlangen, Germany) with a contrast dose of 40 ml and a spatial resolution of 0.6 mm. Finally, 3D reconstructions of mitral valve were obtained using OsiriX MD™ (Pixmeo, Geneva, Switerland) and the mitral valve and LVOT were evaluated.

## Outcomes and definitions

The primary endpoints were 30-day mortality, 1-year mortality, and mortality at follow-up. The secondary endpoint was the development of any complications according to Mitral Valve Academic Research Consortium (MVARC) [6].

## Statistical analysis

Statistical analysis, including regression analysis, was performed using IBM SPSS version 27 (IBM Corp., Chicago, IL, USA) and R software v.3.4.3 (R Foundation for Statistical Computing, Vienna, Austria). Data were tested for normality using the Shapiro-Wilk test. Continuous variables were expressed as medians (interquartile range, IQR) or as mean±standard deviation.

Categorical variables were expressed as frequencies and percentages. We compared the distributions of the categorical variables using Chi-Square Test or Fischer Exact Test if the assumptions for the first one, were not met. The distributions of the continuous variables were compared between the groups with the t-test in cases of normal distributions and with the Mann-Whitney test if the distributions were not normal. Univariate and multivariable logistic regression analyses were performed to identify independent preoperative risk factors for 30-day mortality. Variables identified by the univariate analysis with a P-value <0.05 were included in the multivariable model. Cox proportional hazards regression models were used to determine factors associated with overall survival. The model was verified by the Schoenfeld individual test. A *P*-value of less than 0.05 was considered to indicate statistical significance. For plotting the survival curves and for computing the mid-term mortality we used the Kaplan-Meier method. The cumulative survivals of both methods were analysed and compared with the log rank test.

## Results

### Baseline characteristics

The data on the patients' baseline characteristics are presented in **Table 1.** The mean age at surgery was 69.2±12.2 and male/female ratio was 33/41 (55.4% were female). The SMVR group included 33 patients and the TA-TMVR group consisted of 41 patients. All patients had previously undergone a mitral valve surgery via median sternotomy. The mean time-interval between index surgery and redo surgery (SMVR of TMVR) was 3.62 (IQR 0.74–12.3) years for all patients. The patients in the transcatheter group presented more severe comorbidities, which are reflected by significantly higher risk scores (**Table 1**). Additionally, patients in the transcatheter group had a significantly higher mean systolic pulmonary arterial pressure (63.4 ±16.5 mmHg vs. 49.9±13.0 mmHg, **p<0.001**) and presented with significantly lower left-ventricular ejection fraction 45.5±13.1% vs. 52.2±9.1%, **p = 0.03**).

### Procedure

Intraoperative data is presented in **Table 2**. The overall operating time averaged 148.7 minutes. The mean operating time in the transcatheter group was significantly shorter than in the surgical group (82.8±26.1 vs. 230.6±94.0, **p<0.001**). Also, significantly more patients in the transcatheter group underwent an urgent procedure (**Table 2**). Most of the TA-TMVR procedures were performed under fluoroscopic guidance using a small amount of contrast dye. With our growing experience on the field of transcatheter valve interventions, we have drastically reduced or have completely skipped the use of contrast dye in patients undergoing transcatheter mitral valve interventions. In the present cohort, in 41.5% of the TMVR group the procedure was completely performed with no contrast dye (**Table 1**). All surgical procedures were performed via median re-sternotomy on cardiopulmonary bypass. A total of 5 patients (6.8%) required re-exploration for bleeding (one patient in the transcatheter group) and another three patients (4.1%, all in the SMVR group) suffered from postoperative AV-Block III˚, requiring permanent pacemaker implantation. None of these complications occurred significantly more often in either of the groups. We observed no postoperative myocardial infarction or stroke in our cohort (**Table 3**).

### Survival

Postoperative outcomes' data is presented in **Table 3.** Within the entire patient cohort, the in-hospital, 30-day, and one-year mortality, were 10.8%, 12.2%, and 22.2%, respectively (**Fig 1**).

**Table 1. Baseline characteristics.**

| Characteristics | All patients, %(n) | SMVR, %(n) | TA-TMVI, %(n) | P-value |
|---|---|---|---|---|
| Female gender | 55.4(41) | 66.7(22) | 46.3(19) | 0.1 |
| Age, years | 69.2±12.2 | 63.7±12.8 | 73.6±9.7 | 0.001 |
| Body Mass Index, kg/m$^2$ | 26.4±4.6 | 26.3±4.3 | 26.4±4.8 | 0.59 |
| NYHA III | 58.1(43) | 42.4(14) | 70.7(29) | 0.02 |
| NYHA IV | 29.7(22) | 30.3(10) | 29.3(12) | 1.0 |
| Arterial hypertension | 97.3(72) | 93.9(31) | 100(41) | 0.2 |
| Pulmonary hypertension | 85.1(63) | 66.7(22) | 100(41) | <0.001 |
| Diabetes | 24.3(18) | 12.1(4) | 34.1(14) | 0.03 |
| Chronic obstructive lung disease | 29.7(22) | 15.2(5) | 41.5(17) | 0.02 |
| Coronary artery disease | 48.6(36) | 21.2(7) | 70.7(29) | <0.001 |
| Prior percutaneous coronary intervention | 20.3(15) | 6.1(2) | 31.7(13) | 0.008 |
| Peripheral arterial disease | 21.6(16) | 3.0(1) | 36.6(15) | <0.001 |
| Cerebral arterial disease | 14.9(11) | 6.1(2) | 22.0(9) | 0.1 |
| Prior stroke | 16.2(12) | 24.2(8) | 9.8(4) | 0.1 |
| Sinusrhythmus | 35.1(26) | 39.2(13) | 31.7(13) | 0.62 |
| Atrial fibrillation | 64.9(48) | 60.6(20) | 68.3(28) | 0.62 |
| Aortic regurgitation >II˚ | 5.4(4) | 9.1(3) | 2.4(1) | 0.32 |
| Aortic stenosis >II˚ | 9.5(7) | 6.1(2) | 12.2(5) | 0.45 |
| Mitral regurgitation | 91.9(68) | 87.9(29) | 95.1(39) | 0.4 |
| Mitral regurgitation I˚ | 16.2(12) | 9.1(3) | 22.0(9) | 0.2 |
| Mitral regurgitation II-III˚ | 77.0(57) | 81.8(27) | 73.2(30) | 0.42 |
| Mitral stenosis >II˚ | 45.9(34) | 42.4(14) | 48.8(20) | 0.64 |
| Tricuspid regurgitation <II˚ | 39.2(29) | 39.4(13) | 39.0(16) | 1.0 |
| Tricuspid regurgitation>II˚ | 55.4(41) | 48.5(16) | 61.0(25) | 0.35 |
| Transvalvular mean gradient, mmHg | 9.1±6.7 | 9.7±8.1 | 8.8±5.4 | 0.7 |
| Prior sternotomy | 100(74) | - | - | - |
| Prior mitral valve replacement | 55.4(41) | 48.5(16) | 61.0(25) | 0.35 |
| Prior mitral valve repair | 44.6(33) | 51.5(17) | 39.0(16) | 0.35 |
| Prior coronary artery bypass grafting | 40.5(30) | 15.2(5) | 61.0(25) | <0.001 |
| Prior surgical aortic valve replacement | 18.9(14) | 9.1(3) | 26.8(11) | 0.05 |
| Prior pacemaker implantation | 18.9(14) | 9.1(3) | 26.8(11) | 0.07 |
| Systolic pulmonary arterial pressure, mmHg | 57.4±16.4 | 49.9±13.0 | 63.4±16.5 | <0.001 |
| Ejection fraction, % | 48.5±11.9 | 52.2±9.1 | 45.5±13.1 | 0.03 |
| Chronic kidney injury | 59.5(44) | 48.5(16) | 68.3(28) | 0.1 |
| Dialysis | 9.5(7) | 9.1(3) | 9.8(4) | 1.0 |
| Creatinine, mg/dL | 1.6±1.0 | 1.4±0.9 | 1.7±1.1 | 0.16 |
| GFR (mL/min/1.73m2) | 48.1±19.7 | 52.4±19.6 | 44.6±19.3 | 0.075 |
| Preoperative anticoagulation | | | | |
| Aspirin | 44.6(33) | 21.2(7) | 63.4(26) | <0.001 |
| Clopidogrel | 10.8(8) | 6.1(2) | 14.6(6) | 0.3 |
| Ticagrelor | 1.4(1) | 0 | 2.4(1) | 1.0 |
| Apixaban | 6.8(5) | 0 | 12.2(5) | 0.06 |
| Procoumaron | 36.5(27) | 48.5(16) | 26.8(11) | 0.09 |
| Logistic EuroSCORE, % | 38.0±23.4 | 32.6±25.9 | 42.3±20.5 | 0.018 |
| EuroSCORE II, % | 19.9±16.7 | 18.2±18.9 | 21.2±14.8 | 0.024 |
| STS-Score, % | 11.1±12.5 | 10.2±14.3 | 11.9±10.8 | 0.003 |

*GFR–glomerular filtration rate, NYHA–New-York Heart Association Class, STS Score–Society of Thoracic Surgeons*

**Table 2. Intraoperative characteristics.**

| Characteristics | All patients, %(n) | SMVR, %(n) | TA-TMVI, %(n) | P-value |
|---|---|---|---|---|
| elective | 58.1(43) | 66.7(22) | 51.2(21) | 0.24 |
| urgent | 32.4(24) | 18.2(6) | 43.9(18) | 0.02 |
| emergent | 9.5(7) | 15.2(5) | 4.9(2) | 0.23 |
| Transapical access | - | 0 | 100(41) | - |
| Valve-in-valve | - | 0 | 61(25) | - |
| Valve-in-ring | - | 0 | 39(16) | - |
| Concomitant SAVR | 5.4(4) | 12.1(4) | 0 | 0.04 |
| Concomitant TA-TAVI | 10.8(8) | 0 | 19.5(8) | 0.007 |
| Operating time, min | 148.7±98.6 | 230.6±94.0 | 82.8±26.1 | <0.001 |
| CPB-time, min | - | 138.3±61.7 | - | - |
| Cross-clamp time, min | - | 83.2±42.3 | - | - |
| Constrast dye, mL | - | 0 | 40.0(IQR 0–75.0) | - |
| No contrast dye | 67.6(50) | 44.6(33) | 23(17) | - |
| Fluoroscopy time, min | - | 0 | 14.1±11.8 | - |
| Valve prosthesis size | | 28.6±2.4 | 28.07±1.5 | 0.32 |
| biological prosthesis | 90.5(67) | 78.8(26) | 100(41) | 0.002 |
| mechanical prosthesis | 9.5(7) | 21.2(7) | 0 | 0.002 |

*CPB–cardiopulmonary bypass, SAVR–surgical aortic valve replacement, TA-TAVI–transapical transcatheter aortic valve implantation*

In patients who underwent a surgical redo mitral valve replacement, the in-hospital, 30-day, and one-year mortality was 15.2%, 15.2%, and 18.3%, respectively. In patients with TA-TMVR, the in-hospital, 30-day, and one-year mortality were 7.3%, 9.9%, and 25.4%, respectively (**Table 3**). At one year, mortality did not significantly differ between the groups (p = 0.19) (**Fig 2**). The Kaplan-Meier overall survival curve is given in **Fig 1**. **Fig 3** shows the regression model with significant impact of pulmonary hypertension on 'mid-term´ survival.

## Discussion

In the present study, a total of 74 high-risk patients presenting with failed mitral valve bio-prosthesis or mitral valve annuloplasty rings were treated either by conventional surgical redo mitral valve replacement or a transapical transcatheter mitral valve-in-valve or valve-in-ring implantation. This study provides a number of interesting findings:

1. TA-TMVR is a feasible and uncomplicated treatment option that is at least a non-inferior alternative to conventional surgical redo procedures in selected high-risk candidates.

2. Both methods offer a high technical procedural success, nonetheless, TA-TMVR offers significantly shorter operating times and shorter intensive care unit stay.

3. Although not reaching significance, only SMVR patients needed postoperative pacemaker implantations.

4. Despite the use of contrast medium in the TA-TMVR group, there was no significant difference in postoperative new onset dialysis rate between the groups.

5. There was no significant difference in the hemodynamic performance between the groups. Both methods provided low transvalvular gradients at 'mid-term´ and a low risk of LVOT obstruction.

**Table 3. Postoperative outcomes.**

| Characteristics | All patients, %(n) | SMVR, %(n) | TA-TMVI, %(n) | P-value |
|---|---|---|---|---|
| Paravalvular leakage | 0 | 0 | 0 | - |
| Postoperative mitral regurgitation>trace | 9.5(7) | 0 | 17.1(7) | 0.15 |
| Mean gradient at follow-up | 4.0±1.0 | 3.9±1.2 | 4.2±0.8 | 0.08 |
| Device success | 100% | 100% | 100% | - |
| Dislocation | 0 | 0 | 0 | - |
| Conversion to conventional procedure | - | - | 0 | - |
| New onset atrial fibrillation | 18.9(14) | 27.3(9) | 12.2(5) | 0.13 |
| Acute kidney failure with dialysis | 17.6(13) | 27.3(9) | 9.8(4) | 0.07 |
| Exploration for bleeding | 6.8(5) | 12.1(4) | 2.4(1) | 0.16 |
| Stroke | 0 | 0 | 0 | - |
| Vascular complications | 0 | 0 | 0 | - |
| Pacemaker implantation | 4.1(3) | 9.1(3) | 0 | 0.08 |
| Deep wound infection | 1.4(1) | 3.0(1) | 0 | 0.4 |
| Myocardial infarction | 0 | 0 | 0 | - |
| Re-intubation | 2.7(2) | 3.0(1) | 2.4(1) | 1.0 |
| Shock | 14.9(11) | 21.2(7) | 9.7(4) | 0.5 |
| cardiogenic | 9.5(7) | 12.1(4) | 7.3(3) | 0.7 |
| septic | 5.4(4) | 9.1(3) | 2.4(1) | 0.3 |
| Time on respirator, days | 1.0(IQR 1.0–3.1) | 1.0(1.4–4.3) | 1.0(IQR 0.55–2.4) | 0.21 |
| Time on ICU, days | 2.0(IQR 1.0–5.2) | 4.0(IQR 3.6–6.3) | 2.0(IQR 1.6–4.6) | <0.001 |
| In-hospital stay, days | 9.0(IQR 7.0–13.25) | 11.0 (IQR 8.4–18.4) | 9.7±5.4 | 0.06 |
| In-hospital mortality | 10.8(8) | 15.2(5) | 7.3(3) | 0.45 |
| 30-day mortality | 12.2(9) | 15.2(5) | 9.8(4) | 0.501 |
| Follow-up time, days | 997.3(IQR 203.25–1443.25) | 1163.0(IQR 928.5–1733.5) | 728.6(IQR 510.6–946.6.0) | 0.052 |
| 1-year mortality | 22.2 | 18.3 | 25.4 | 0.19 |
| 3-year mortality | 27.1 | 27.1 | 37.4 | - |

*ICU–Intensive Care Unit*

6. There was no significant difference in 30-days and one-year mortality.

Following the great success of transcatheter technologies in aortic valve replacement, transcatheter mitral valve-in-valve (TMViV) or valve-in-ring (TMViR) implantation has also recently been rapidly developing as an alternative to conventional surgical MV redo procedures. Therefore, we sought to evaluate our results with transapical transcatheter mitral valve replacement in high-risk patients with prohibitive surgical risk and to compare them to the conventional redo SMVR. The observational period in the present study was 8 years.

In the transcatheter group, patients were significantly older and presented more comorbidities, which is reflected by higher EuroSCORE-II and STS-Scores compared to the surgical group. Nonetheless, there was no difference in the survival between the groups. While redo SMVR is known to carry a high periprocedural and postoperative mortality risk, mortality rates of the present study are in line with those reported in previous studies [2, 7, 8]. A study published by Kamioka et al. evaluating the results of 59 patients undergoing either TMVR of SMVR reported no significant difference in the survival of both procedures [9], which was proven by our results.

It is our philosophy to replace the mitral valve during reoperation, if re-repair prognostic success was deemed low, which is common sense: Trumello et al. suggested that, as far as long-

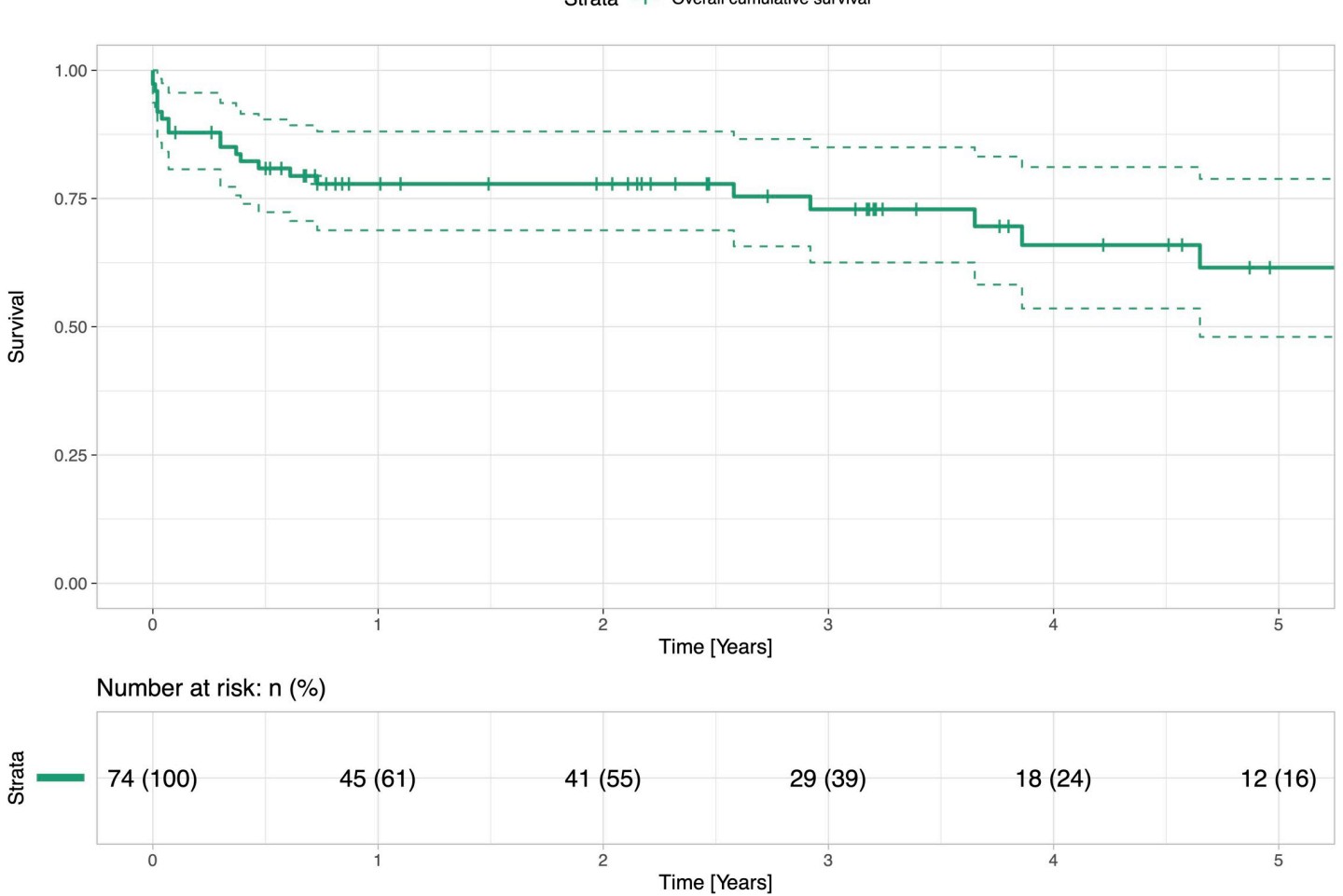

**Fig 1. Overall survival.** The figure shows overall survival of all patients from the cohort presented with a Kaplan- Meier Curve.

term durability is concerned, re-repair of the failed mitral annuloplasty ring should only be pursued in cases where the intraoperative findings and immediate results were very reassuring [10]. In the case of TMVR within a failed annuloplasty ring, obstruction of the LVOT might be a concern [11]. Regueiro et al. clearly showed, that the expansion of the transcatheter valve prosthesis could result in displacement of the anterior mitral leaflet into the LVOT leading to potential obstruction [12]. The incidence of the LVOT obstruction after TMVR is reported to be higher than after redo SMVR [13, 14]. Therefore, extensive CT-based evaluation of the LVOT prior to TMVR is essential. By doing so, neither in our transcatheter arm, nor in our surgical group, any relevant LVOT obstruction occurred. In our study, LVOT was defined according to the MVARC criteria described by Stone et al [15].

The current guidelines on valvular heart disease recommend concomitant tricuspid valve (TV) repair in patients presenting with more than moderate tricuspid valve regurgitation [16]. As this concomitant procedure is known not to influence peri-procedural mortality [17, 18], we did not exclude concomitant TV procedures from the present analysis. However, by nature, only in the SMVR group concomitant tricuspid valve repair was performed. Although current guidelines recommend concomitant TV repair, long-term or mid-term outcomes of concomitant TV repair in redo cases remains controversial. Kamioka et al. reported no significant

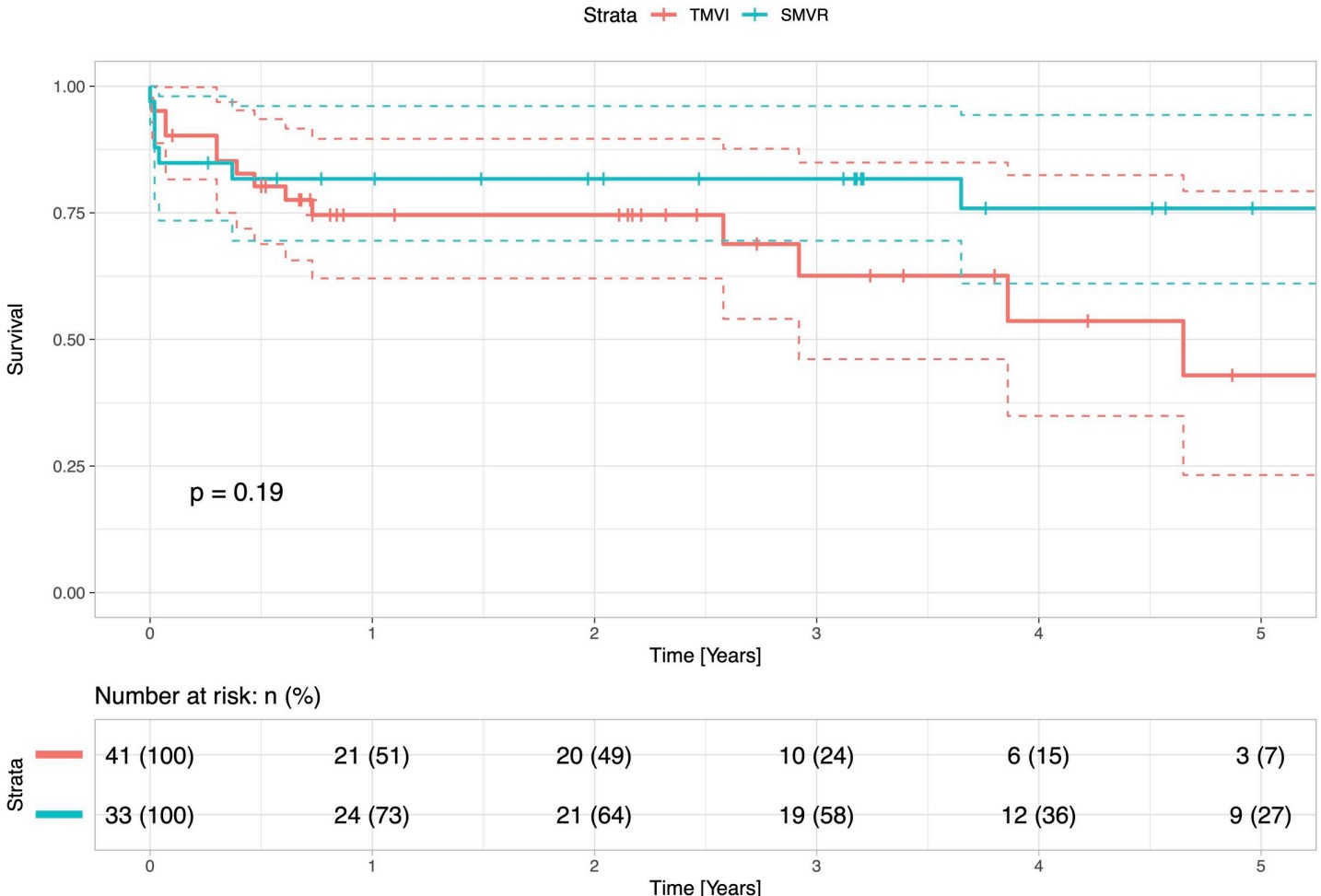

**Fig 2. Survival of patients undergoing redo SMVR and TA-TMVR.** The figure presents the survivals of patients undergoing a redo SMVR and TA-TMVR presented with Kaplan-Meier curves. The survival rates have been analysed and compared with the log rank test and show no statistical difference (p>0.05).

difference in one-year mortality in patients undergoing redo SMVR with or without TV repair [9]. Similarly, we did not discover any significant difference in the mortality between the groups, even though 39.4% (n = 13) of patients in the surgical arm received concomitant TV repair. The rate of more than moderate preoperative tricuspid valve regurgitation was similar between the groups. As the current analysis reflects a 5-year observation period, the "late" impact of tricuspid regurgitation might be underestimated. Nevertheless, we could show, that severe pulmonary hypertension, as an indirect parameter of right heart function, has a significant impact at least on mid-term survival (P = 0.043) as shown in **Fig 3**.

In our cohort, we found no significant difference in postoperative major adverse events between the surgical and transcatheter cohort. Kidney function is known to be an independent predictive factor of mortality in patients undergoing transcatheter aortic valve procedures [19, 20]. The correlation between the contrast dye dose used and post-procedural acute kidney injury was reported by Yamamoto et al. [21]. Although, most of the TA-TMVR in our cohort were performed with a very small amount of contrast dye or even none at all, no significant difference was noted in postoperative new onset dialysis between the groups. Similar outcomes have also been reported by Simonetto et al. [22]. We believe that thanks to the fluoroscopic

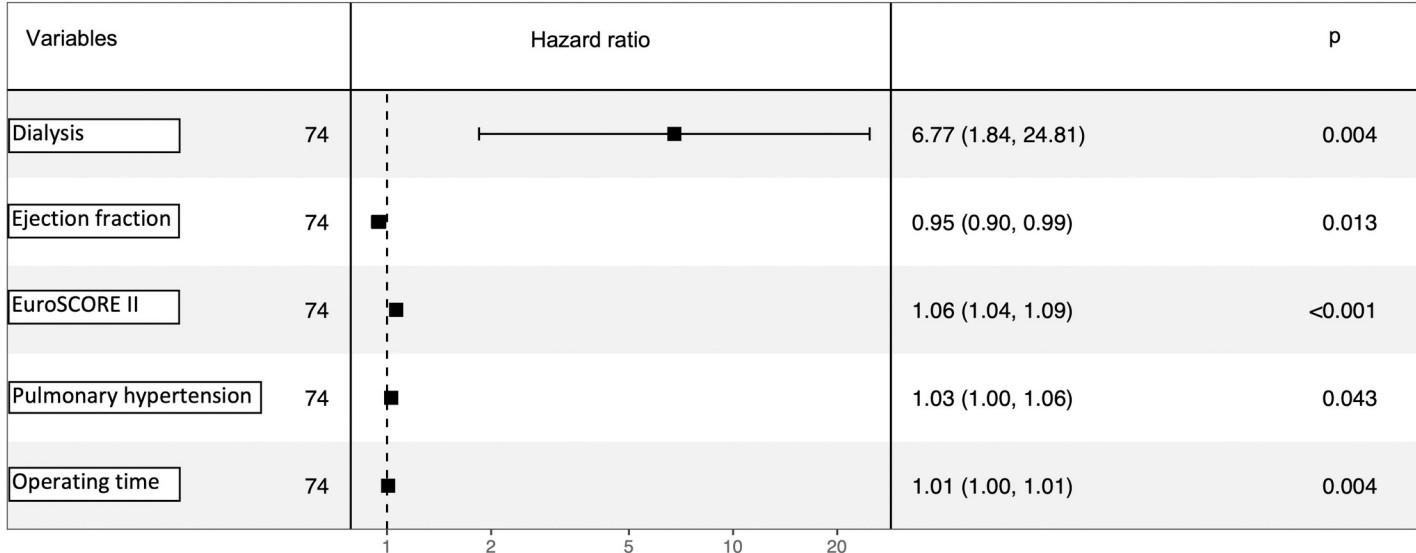

| Variables | | Hazard ratio | p |
|---|---|---|---|
| Dialysis | 74 | 6.77 (1.84, 24.81) | 0.004 |
| Ejection fraction | 74 | 0.95 (0.90, 0.99) | 0.013 |
| EuroSCORE II | 74 | 1.06 (1.04, 1.09) | <0.001 |
| Pulmonary hypertension | 74 | 1.03 (1.00, 1.06) | 0.043 |
| Operating time | 74 | 1.01 (1.00, 1.01) | 0.004 |

**Fig 3. Cox-regression analysis.**

qualities of prior implanted surgical mitral valve prostheses and mitral rings, the amount of contrast dye could be further reduced or completely excluded [5].

Although, a high proportion of our patients underwent a valve-in-ring procedure, we observed no specific valve related complications such as valve embolization of LVOT obstruction. It is known, that patients undergoing transcatheter mitral valve-in-ring procedures might present with 'paravalvular´ leakage between the transcatheter heart valve and the annuloplasty ring [23]. This was not the case in the present analysis. A small number of patients (17%) showed mild mitral regurgitation, coming from a small central leakage of the SAPIEN prosthesis. As the transcatheter heart valve during TMVR is implanted in a pre-shaped 'docking´-station of known size (**Fig 4**), the risk of postoperative pacemaker implantation in inherently lower. Although not reaching statistical significance, none of the patients of the transcatheter group needed postoperative pacemaker implantation.

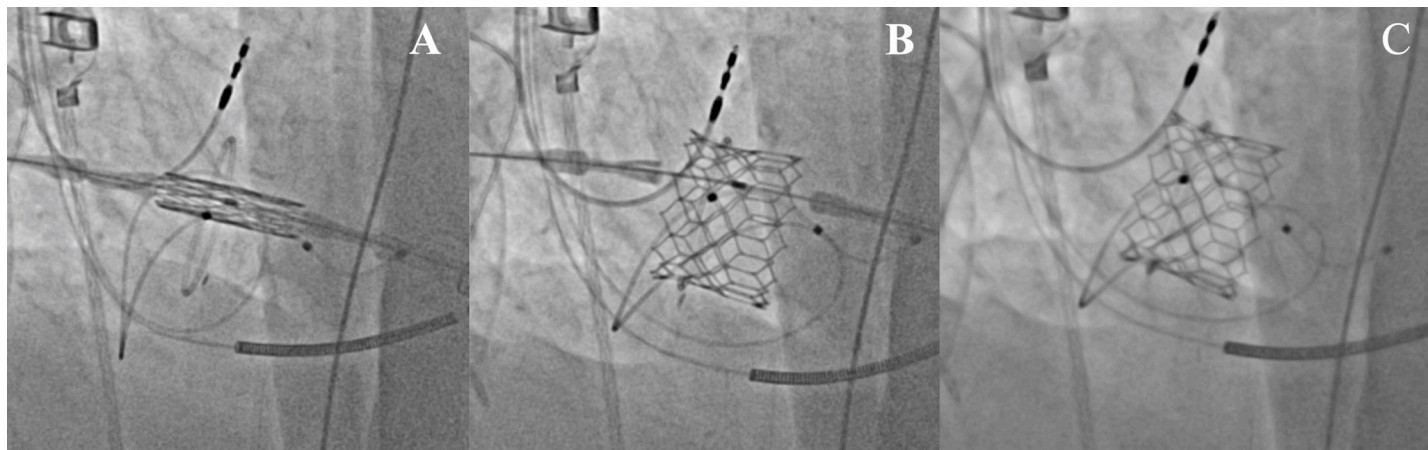

**Fig 4. Transapical transcatheter mitral valve implantation valve-in-valve in a pre-shaped 'docking´-station of known size.** A–Positioning of the valve prosthesis in the annuloplasty ring. B–Deployment of the transcatheter valve prosthesis in the mitral position. C–Fully deployed valve prosthesis in the annuloplasty ring.

In accordance with the findings of previous studies, we discovered that both procedures provide comparable and excellent hemodynamic results with low transvalvular gradients at follow-up [9, 11, 22]. Follow-up echocardiography showed mean transvalvular gradients in the transcatheter arm of 4.2±0.8mmHg, compared to 3.9±1.2 mmHg in the surgical group.

All COPD patients within the present analysis were on all inhalative bronchodilatator therapy preoperatively. Our institutional concept is, that patients presenting with a FEV1> 1L were deemed to be operable, for both the transcatheter or conventional surgical approach. Nevertheless, we modified our anesthetic strategy for the transapical approach in the end of 2019: for the transapical approach, we moved from endotracheal intubation towards the use of a laryngeal mask, and only short-acting opioids (remifentanil) were used. With this concept, all patients could be transferred spontaneously breathing to the ICU. All surgical treated patients received an additional epidural anesthetic support.

Despite all the benefits of transcatheter mitral valve procedures, there are several conditions which make the patients ineligible for this therapeutic option. Patients presenting with infective endocarditis, failed annuloplasty treated with an open annuloplasty band or closed rings larger than 34mm present a problem for transcatheter valve replacement and should undergo conventional redo surgery. Moreover, long-term durability of transcatheter valves are still under investigation especially in the mitral valve-in-valve or valve-in-ring position.

## Conclusion

Whilst surgical redo mitral valve replacement remains the gold standard of care in patients presenting with degenerated biological mitral valve prostheses or failed mitral valve annuloplasty, transcatheter options have been rapidly evolving as a valid alternative, especially in patients presenting with high operative risk. Although, there are several contraindications for transcatheter mitral valve replacement, in our study we demonstrated the non-inferiority of the transapical transcatheter mitral valve replacement compared to conventional surgical redo procedures in a high-risk cohort. TA-TMVR offers excellent hemodynamic results and similar survival compared to the conventional surgical method. Although not significant, early mortality is nearly doubled in the SMVR group. Additionally, TA-TMVR comes with shorter procedure duration and shorter ICU stay, which also lowers the costs of the procedure. Table 4 summarizes the potential inclusion or exclusion criteria to guide decision-making for TA-TMVR.

## Study limitations

The retrospective non-randomized nature of the study coming from a single center with a limited number of patients may have an impact on the outcomes and the study power, and can leave room for bias. So far, only few studies with smaller single-center cohorts on this topic

**Table 4. Inclusion / exclusion criteria for the transapical TMVinV/TMVinR approach.**

| Inclusion | Exclusion |
|---|---|
| **eligibility for transapical access** | mechanical mitral valve prosthesis |
| **prohibitive risk for the conventional surgical approach** | hostile anatomy of the left thorax |
| **"heart-team" decision** | thrombus within the left apex |
| **patients wish** | Large annuloplasty rings (>34mm) |
| **Pre-existing ASD Amplatzer device with no possibility for the transseptal approach** | High-risk for LVOT obstruction |

have been published. Further prospective studies on larger cohorts should be conducted to validate the safety and efficiency of this method.

## Supporting information

**S1 File.**
(XLSX)

## Author Contributions

**Conceptualization:** Alina Zubarevich, Matthias Thielmann, Bastian Schmack, Arjang Ruhparwar, Alexander Weymann, Daniel Wendt.

**Data curation:** Alina Zubarevich, Marcin Szczechowicz, Arian Arjomandi Rad, Robert Vardanyan, Philipp Marx, Alexander Lind, Rolf Alexander Jánosi, Mehdy Roosta-Azad, Rizwan Malik, Bastian Schmack.

**Formal analysis:** Alina Zubarevich, Marcin Szczechowicz, Rizwan Malik, Markus Kamler, Matthias Thielmann, Mohamed El Gabry, Bastian Schmack.

**Funding acquisition:** Daniel Wendt.

**Investigation:** Alina Zubarevich, Marcin Szczechowicz, Philipp Marx.

**Methodology:** Alina Zubarevich, Markus Kamler, Arjang Ruhparwar.

**Project administration:** Alexander Lind, Rolf Alexander Jánosi, Mehdy Roosta-Azad, Alexander Weymann, Daniel Wendt.

**Software:** Marcin Szczechowicz.

**Supervision:** Arjang Ruhparwar, Alexander Weymann, Daniel Wendt.

**Writing – original draft:** Alina Zubarevich.

**Writing – review & editing:** Alina Zubarevich, Marcin Szczechowicz, Arian Arjomandi Rad, Robert Vardanyan, Philipp Marx, Alexander Lind, Rolf Alexander Jánosi, Mehdy Roosta-Azad, Rizwan Malik, Markus Kamler, Matthias Thielmann, Mohamed El Gabry, Bastian Schmack, Arjang Ruhparwar, Alexander Weymann, Daniel Wendt.

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
