## [Decision Letter · Decision Letter 0]

1 Jul 2021

PONE-D-21-18022

Mitral Surgical Redo versus Transapical Transcatheter Mitral Valve Implantation

PLOS ONE

Dear Dr. Zubarevich,

Thank you for submitting your manuscript to PLOS ONE. After careful consideration, we feel that it has merit but does not fully meet PLOS ONE’s publication criteria as it currently stands. Therefore, we invite you to submit a revised version of the manuscript that addresses the points raised during the review process.

We look forward to receiving your revised manuscript.

Kind regards,

Alessandro Parolari, MD, PhD

Academic Editor

PLOS ONE

Journal Requirements:

1. Please ensure that your manuscript meets PLOS ONE's style requirements, including those for file naming. The PLOS ONE style templates can be found athttps://journals.plos.org/plosone/s/file?id=wjVg/PLOSOne_formatting_sample_main_body.pdf and https://journals.plos.org/plosone/s/file?id=ba62/PLOSOne_formatting_sample_title_authors_affiliations.pdf

'Daniel Wendt is working as a proctor for Edwards Lifesciences. Other authors have no

conflicts of interest.'

Additional Editor Comments (if provided):

Reviewers' comments:

Reviewer's Responses to Questions

**Comments to the Author**

1. Is the manuscript technically sound, and do the data support the conclusions?

Reviewer #1: Yes

Reviewer #2: Yes

2. Has the statistical analysis been performed appropriately and rigorously? 

Reviewer #1: Yes

Reviewer #2: No

3. Have the authors made all data underlying the findings in their manuscript fully available?

Reviewer #1: Yes

Reviewer #2: Yes

4. Is the manuscript presented in an intelligible fashion and written in standard English?

Reviewer #1: Yes

Reviewer #2: Yes

5. Review Comments to the Author

Reviewer #1: This is a mono-centric, retrospective study. The Authors described their experience about redo mitral valve treatment in a period of 8 years. They compared 74 patients underwent to mitral valve replacement or a mitral valve repair that come back for either with dysfunctional biological mitral valve prosthesis or with failed ring-annuloplasty; (33 pt-> SMVR; 41 pt TA-TMVR).

The analysis, and after the conclusion, clearly showed, in this center, the non-inferiority of transapical mitral valvular replacement compared to convention surgical redo procedure.

Please explain:

Table 2

Operating time: what do you mean? In surgical approach usually we may distingue cross clamp time and CBP time

Table 3

A total of 9 patients had shock: 7 in MVR group and 2 in TAMVR; tot 9

7 patient had cardiogenic shock: 4 in MVR and 3 in TAMVR; tot 7

4 patient had a septic shock: 3 in MVR and 1 in TAMVR; tot 4

7 patient (cardiogenic shock)+ 4 patient (septic shock) ; tot 11????? You describe a total of 9 patient with shock;

2 patient in TAMVR but 3+1-> 4

My impression about this manuscript is that the colleagues write a good paper, they correctly described the center experience and critically analyze the results.

Reviewer #2: I read with great interest the paper of the Essen group, comparing surgery and TMVR in a cohort of patients with failing MV procedures. The authors collected a consecutive series of 74 patients, which is a quite huge population and they are to be congratulated for their excellent results. I have some comments that, in my opinion, should be addressed to better clarify the indications and strengthen the manuscript’s message:

1) Patients in TMVR group have significant incidence of comorbidities compared to the MVR group. Among them, COPD is present in 40% of patients. Generally, severe COPD is a risk factor for open surgery, but it is also a contraindication for left thoracotomy (even if with small incision). Among this 40% of COPD patients, how many presented with inhalators therapy or severe pulmonary dysfunction? In the authors’ experience, which are the pulmonary or LV dysfunction condition that are considered high risk or frank contra-indication for TA-TMVR?

2) In TMVR group, 70% of patients presented with CAD, even if only 31% with previous PCI. In the other 40% of patients, was CAD a chronic situation without any indication for concomitant myocardial revascularization? In your practice, the concomitant need of myocardial revascularization influences the decision in favor of MVR vs TMVR? In case of TMVR plus PCI, which is your strategy? A concomitant procedure in hybrid suit or a staged one?

3) I am not sure that the issue of TR is to be considered irrelevant. Several surgical literature clearly showed that even moderate TR secondary to MV pathology, especially in the presence of pulmonary hypertension, should be aggressively addressed. You stated that you did not see any difference in the follow-up in both group, according to TR repair. However, I think that three years of FU is a limited time-frame to assess TR impact. Furthermore (I’ll discuss this issue later) a multivariable analysis, including TV repair, should be performed on early and late mortality, to really assess the prognostic value of any preoperative or intraoperative risk factor.

4) In nearly 50% of both group a MV replacement (or a Mitral ViV) was performed, I presume for bioprosthesis failure. I think that an important missing data is the time between previous surgery and MVR/TMVR, or, in other world, the duration of the previously implanted bioprostheses. Which is the age criteria for bioprosthesis implant according to the authors?

5) In the postoperative data, mild MR is reported in 17% of TMVR patients, but without evident PVL. Which is the mechanism of residual MR in the authors’ opinion?

6) If I look to the risk profile of both population, all patients presented with high risk for surgery (STS 10% vs 12%), even if TMVR patients are older and with more comorbidities. Which are the selection criteria used by the authors in proposing redo surgery vs TA TMVR? Perhaps a brief algorithm or a brief list of inclusion/exclusion criteria would be extremely useful

7) Finally, even if mortality is not statistically significant, there are in my mind two clear facts: early mortality is 15% vs 7-9% in surgery Vs TMVR respectively, which is nearly double anyway, despite patients in TMVR group are more risky. On the contrary at 1-3 years, mortality in TMVR group is higher than the surgical group. In order to better clarify the role of the procedural choice and the impact of the comorbidities, a multivariable analysis for early mortality (logistic regression) and late mortality (Cox regression) should be performed.

6. PLOS authors have the option to publish the peer review history of their article (what does this mean?). If published, this will include your full peer review and any attached files.

Reviewer #1: No

Reviewer #2: No

---

## [Author Response · Author response to Decision Letter 0]

9 Jul 2021

Dr. Emily Chenette

Editor-in-Chief 

PLOS ONE 

 Essen, July 9th 2021 

Dear Dr. Chenette,

Please find enclosed our revised manuscript ID PONE-D-21-18022 entitled: “Mitral Surgical Redo versus Transapical Transcatheter Mitral Valve-in-Valve and Valve-in-Ring Implantation” originally submitted for the category “Research Article” to PLOS ONE.

The authors would like to thank you for your interest in publishing the article in your journal. We have considered all reviewers’ remarks and hope that we could sufficiently improve the article in light of the comments. We also provided a point-to-point answer to the reviews (see below). We highlighted all changes in the revised manuscript with a yellow color.

Yours sincerely and respectfully,

Alina Zubarevich, MD

West German Heart and Vascular Center, Essen/ Germany

Reviewer Comments

Reviewer 1

This is a mono-centric, retrospective study. The Authors described their experience about redo mitral valve treatment in a period of 8 years. They compared 74 patients underwent to mitral valve replacement or a mitral valve repair that come back for either with dysfunctional biological mitral valve prosthesis or with failed ring-annuloplasty; (33 pt->SMVR;41ptTA-TMVR).

The analysis, and after the conclusion, clearly showed, in this center, the non-inferiority of transapical mitral valvular replacement compared to convention surgical redo procedure.

Comment 1:

Table2

Operating time: what do you mean? In surgical approach usually we may distingue cross clamp time and CBP time

My impression about this manuscript is that the colleagues write a good paper, they correctly described the center experience and critically analyze the results.

Reply:

Thank you for your valid comment. In our manuscript, “Operating time” is equal to “Incision to skin closure” time. As the transcatheter mitral valve implantation naturally has no CPB and Cross-Clamp time, we decided to compare both techniques only by the operating time. Nevertheless, we calculated the mean CPB- and ACC-time for the surgical group, which was a CPB-time of 138.3±61.7 minutes and mean Cross-clamp-time was 83.2±42.3 minutes, respectively. We added these values into the Table 2.

Comment 2: 

A total of 9 patients had shock: 7 in MVR group and 2 in TAMVR; tot 9

7 patient had cardiogenic shock: 4 in MVR and 3 in TAMVR; tot 7

4 patient had a septic shock: 3 in MVR and 1 in TAMVR; tot 4

7 patient (cardiogenic shock)+ 4 patient (septic shock) ; tot 11????? You describe a total of 9 patient with shock;

2 patient in TAMVR but 3+1-> 4

Reply:

Thank you for your comment. You are right about the inconsistency of the numbers. There was a typing error in Table 3, which is now corrected. 

Changes in text: 

Table 3, the changes are highlighted in color. 

Reviewer 2

I read with great interest the paper of the Essen group, comparing surgery and TMVR in a cohort of patients with failing MV procedures. The authors collected a consecutive series of 74 patients, which is a quite huge population and they are to be congratulated for their excellent results. I have some comments that, in my opinion, should be addressed to better clarify the indications and strengthen the manuscript’s message:

Comment 1: 

Patients in TMVR group have significant incidence of comorbidities compared to the MVR group. Among them, COPD is present in 40% of patients. Generally, severe COPD is a risk factor for open surgery, but it is also a contraindication for left thoracotomy (even if with small incision). Among this 40% of COPD patients, how many presented with inhalators therapy or severe pulmonary dysfunction? In the authors’ experience, which are the pulmonary or LV dysfunction condition that are considered high risk or frank contra-indication for TA-TMVR?

Reply:

We are grateful for this remark. First, Essen is located within the so called “Ruhrgebiet”, which was famous in the past for coal-mining (during the 70s/80s the largest coal-mining area in Europe), and therefore, still many patients present with COPD or other lung diseases. This explains indeed, the high incidence of COPD in our cohort. All COPD patients within the present analysis were on inhalative bronchodilatator therapy. Our institutional concept is, that patients presenting with a FEV1> 1L were deemed to be operable, for both the transcatheter or conventional surgical approach. However, we modified our anesthetic strategy for the transapical approach in the end of 2019: for the transapical approach, we moved from endotracheal intubation towards the use of a laryngeal mask, and only short-acting opioids (remifentanil) were used. With this concept, all patients could be transferred spontaneously breathing on the ICU. In our experience, COPD or severe lung disease would not count for exclusion for the transapical approach. Moreover, as even in the transpical TMViV or TMViR situation, it still represents a redo-situation and in most cases, only the apex was exposed without opening of the left pleural space.

The following was therefore added in to the revised manuscript:

All COPD patients within the present analysis were on all inhalative bronchodilatator therapy preoperatively. Our institutional concept is, that patients presenting with a FiV1> 1L were deemed to be operable, for both the transcatheter or conventional surgical approach. Nevertheless, we modified our anesthetic strategy for the transapical approach in the end of 2019: for the transapical approach, we moved from endotracheal intubation towards the use of a laryngeal mask, and only short-acting opioids (remifentanil) were used. With this concept, all patients could be transferred spontaneously breathing on the ICU. All surgical treated patients received an additional epidural anesthetic support.

Comment 2: 

In TMVR group, 70% of patients presented with CAD, even if only 31% with previous PCI. In the other 40% of patients, was CAD a chronic situation without any indication for concomitant myocardial revascularization? In your practice, the concomitant need of myocardial revascularization influences the decision in favor of MVR vs TMVR? In case of TMVR plus PCI, which is your strategy? A concomitant procedure in hybrid suit or a staged one?

Reply:

Thank you for your excellent comment. Concomitant CAD in patients presenting with a recurrent valve pathology is very common. In our cohort 40% of patients, who did not undergo a PCI, did not require coronary revascularization, but present with non-significant CAD.

In our daily practice, if the patient is eligible for a conventional procedure and requires concomitant coronary revascularization as defined by the guidelines, CABG will be definitively revascularization being performed simultaneously. 

If the patient is planned for the transcatheter approach, each patient will be discussed in our institutional Heart Team according to each patients’ specific needs including CAD. Our group has already analyzed patients undergoing TAVR presenting with concomitant CAD (Wendt D, Kahlert P, Lenze T, et al. Management of high-risk patients with aortic stenosis and coronary artery disease. Ann Thorac Surg. 2013;95(2):599-605. doi:10.1016/j.athoracsur.2012.07.075.). Within this previous publication, we could show, that patients who were treated by PCI prior to TAVR had similar results in a propensity score adjusted analysis compared to patients treated by AVR and CABG. Therefore, severe CAD should be treated by PCI prior to TAVR. On the other hand, we also know that concomitant CAD and staged revascularization is a significant mortality predictor in patients undergoing TAVR. Therefore and according to the above-mentioned study, we try to treat all significant CAD prior to each interventional approach (at least 4 weeks prior to TAVR intervention).

On the other hand, in some pathologies, were we aim to perform a transaortic TAVR, and during this procedure (in contrast to the transapical TMVinV / TMVinR), CAD can easily be treated by the addition of an OPCAB-procedure. This concept has been just recently published by our group (Zubarevich A, Zhigalov K, Szczechowicz M, et al. Simultaneous transaortic transcatheter aortic valve implantation and off-pump coronary artery bypass: An effective hybrid approach. J Card Surg. 2021;36(4):1226-1231. doi:10.1111/jocs.15351). 

We hope, this answers your question and if wanted, we can add a short comment into the discussion part. 

Comment 3: 

I am not sure that the issue of TR is to be considered irrelevant. Several surgical literatures clearly showed that even moderate TR secondary to MV pathology, especially in the presence of pulmonary hypertension, should be aggressively addressed. You stated that you did not see any difference in the follow-up in both groups, according to TR repair. However, I think that three years of FU is a limited time-frame to assess TR impact. Furthermore (I’ll discuss this issue later) a multivariable analysis, including TV repair, should be performed on early and late mortality, to really assess the prognostic value of any preoperative or intraoperative risk factor.

Reply: 

Thank you for your well-taken comment. Indeed, TR reflects a major factor affecting survival and especially in the long-term. In our present analysis, we evaluated “mid-term” survival over a 5 years period, which is indeed a limited time-frame. As suggested by the reviewer, we went back into our dataset and we constructed (as requested) a regression analysis. Within the univariate analysis, we evaluated TR (I-II°), TR III-III+°concomitant TR surgery and severe pulmonary hypertension in regard to 30-day mortality, but none of these parameters showed any significant impact. 

TR I-II°: OR 0.402 (95% CI 0.056-1.818), P=0.278

TR III-III+°: OR 1.714 (95% CI 0.414-8.673), P=0.472

Concomitant TR-surgery: OR 1.402 (95% CI 0.191-6.800), P=0.696

sPAP: OR 1.021 (95% CI 0.979-1.062), P=0.294

In a next step, we performed the same regression analysis in regard to “mid-term” survival for the COX-regression analysis. This model also showed no significant impact the tricuspid valve. TR could not be included into the final model, as the proportional Hazard assumption of the model could not be full-filled. Nevertheless, as the reviewer requested, we constructed the model (without TR) and in this model, at least severe pulmonary hypertension, as an indirect parameter of right heart function showed a significant impact on “mid-term” survival. 

TR I-II°: HR 0.896 (95% CI 0.376-2.132), P=0.804

TR III-III+°: HR 1.377 (95% CI 0.595-3.186), P=0.454

Concomitant TR-surgery: HR 0.820 (95% CI 0.277-2.424), P=0.720

sPAP: HR 1.024 (95% CI 1.001-1.047), P=0.04

This final regression model fulfilled the Hazard proportion assumption as was verified by the Schoenfeld residual test with an AUC of 0.94. 

Therefore, the following part was added into the stastistical section of the material and methods part:

Univariate and multivariable logistic regression analyses were performed to identify independent preoperative risk factors for 30-day mortality. Variables identified by the univariate analysis with a P-value <0.05 were included in the multivariable model. Cox proportional hazards regression models were used to determine factors associated with overall survival. The model was verified by the Schoenfeld individual test. 

Another figure #3 with a corresponding text passage were added into the results part, displaying the results of the COX regression analysis, with emphasis on pulmonary hypertension:

Figure 3 shows the regression model with significant impact of pulmonary hypertension on `mid-term´ survival.

Moreover, these new results were also discussed in the revised discussion part:

As the current analysis reflects a 5-year observation period, the “late” impact of tricuspid regurgitation might be underestimated. Nevertheless, we could show, that severe pulmonary hypertension, as an indirect parameter of right heart function, has a significant impact at least on mid-term survival.

Comment 4: 

In nearly 50% of both group a MV replacement (or a Mitral ViV) was performed, I presume for bioprosthesis failure. I think that an important missing data is the time between previous surgery and MVR/TMVR, or, in other world, the duration of the previously implanted bioprostheses. Which is the age criteria for bioprosthesis implant according to the authors?

Reply:

Thank you for your valid comment. First of all, in our center, the percentage of mitral valve repair is >90%. Most of the included patients were operated elsewhere, and were referred to our center for redo surgery, whether due to failed mitral valve repair or early-generated mitral valve bioprosthesis. As asked by the reviewer, the median time between previous surgery and MVR/TMVR was 3.62(IQR 0.74-12.3) years. Our institutional age criteria are reflected by the current guidelines, where mechanical mitral valve prostheses should be considered for patients younger than 60-65 years, if these is no contraindication for warfarin therapy. Nevertheless, individual patients’ comorbidities and the ability to manage warfarin-therapy must be taken into consideration. Moreover, as the transcatheter mitral valve implantation techniques are rapidly developing, biological valve implantation in the mitral position does not necessarily mean a risky re-operation anymore and is at least in Germany steadily increasing.

We added the following into the revised results part:

The mean time-interval between index surgery and redo surgery (SMVR of TMVR) was 3.62 (IQR 0.74-12.3) years for all patients.

Comment 5: 

In the postoperative data, mild MR is reported in 17% of TMVR patients, but without evident PVL. Which is the mechanism of residual MR in the authors’ opinion?

Reply:

We are thankful for this remark. Indeed, the residual MR in these patients comes from a small proportion of central leakage coming from the SAPIEN prosthesis.

The following was added into the revised discussion:

A small number of patients (17%) showed mild mitral regurgitation, coming from a small central leakage of the SAPIEN prosthesis.

Comment 6: 

If I look to the risk profile of both populations, all patients presented with high risk for surgery (STS 10% vs 12%), even if TMVR patients are older and with more comorbidities. Which are the selection criteria used by the authors in proposing redo surgery vs TA TMVR? Perhaps a brief algorithm or a brief list of inclusion/exclusion criteria would be extremely useful

Reply:

Thank you for your well-taken comment. The high-risk nature of the redo procedure itself represents the main driving factor to go for an interventional approach. In detail, if the patient comes with several comorbidities, requires no further concomitant procedures and is technical feasible (e.g. annuloplasty ring max. 34mm, biological prosthesis in the mitral position) we would aim to treat the patient interventionally. Moreover, all decisions were made by the interdisciplinary institutional “heart-team”. 

We like your suggestion to implement a short inclusion/exclusion manual, and therefore added the following table into the revised manuscript, to give the reader some useful tips.

Inclusion Exclusion

eligibility for transapical access mechanical mitral valve prosthesis

prohibitive risk for the conventional surgical approach hostile anatomy of the left thorax

“heart-team” decision thrombus within the left apex

patients wish Large annuloplasty rings (>34mm)

Pre-existing ASD Amplatzer device with no possibility for the transseptal approach High-risk for LVOT obstruction

Comment 7: 

Finally, even if mortality is not statistically significant, there are in my mind two clear facts: early mortality is 15% vs 7-9% in surgery Vs TMVR respectively, which is nearly double anyway, despite patients in TMVR group are riskier. On the contrary at 1-3 years, mortality in TMVR group is higher than the surgical group. In order to better clarify the role of the procedural choice and the impact of the comorbidities, a multivariable analysis for early mortality (logistic regression) and late mortality (Cox regression) should be performed.

Reply:

Thank you for your comment, which was answered partly above about the regression analysis. We totally agree with your excellent comment, that early mortality is nearly doubled (although not significant) in the SMVR group. By nature, as the TMVR patients are older and showed more comorbidities (reflected by higher risk-scores), long-term/mid-term survival of the interventional group is worse compared to the younger SMVR group. 

Therefore, to emphasis this important point, the following was added into the conclusion part:

Although not significant, early mortality is nearly doubled in the SMVR group.

---

## [Decision Letter · Decision Letter 1]

10 Aug 2021

Mitral Surgical Redo versus Transapical Transcatheter Mitral Valve Implantation

PONE-D-21-18022R1

Dear Dr. Zubarevich,

We’re pleased to inform you that your manuscript has been judged scientifically suitable for publication and will be formally accepted for publication once it meets all outstanding technical requirements.

Kind regards,

Alessandro Parolari, MD, PhD

Academic Editor

PLOS ONE

Additional Editor Comments (optional):

Reviewers' comments:

Reviewer's Responses to Questions

**Comments to the Author**

1. If the authors have adequately addressed your comments raised in a previous round of review and you feel that this manuscript is now acceptable for publication, you may indicate that here to bypass the “Comments to the Author” section, enter your conflict of interest statement in the “Confidential to Editor” section, and submit your "Accept" recommendation.

Reviewer #1: All comments have been addressed

Reviewer #2: All comments have been addressed

2. Is the manuscript technically sound, and do the data support the conclusions?

Reviewer #1: Yes

Reviewer #2: Yes

3. Has the statistical analysis been performed appropriately and rigorously? 

Reviewer #1: Yes

Reviewer #2: Yes

4. Have the authors made all data underlying the findings in their manuscript fully available?

Reviewer #1: Yes

Reviewer #2: Yes

5. Is the manuscript presented in an intelligible fashion and written in standard English?

Reviewer #1: Yes

Reviewer #2: Yes

6. Review Comments to the Author

Reviewer #1: The authors considered all suggestions and corrected the manuscript and table where necessary. This is a good manuscript to explain their experience

Reviewer #2: (No Response)

7. PLOS authors have the option to publish the peer review history of their article (what does this mean?). If published, this will include your full peer review and any attached files.

Reviewer #1: No

Reviewer #2: **Yes: **Andrea Garatti

---

## [Editor Report · Acceptance letter]

16 Aug 2021

PONE-D-21-18022R1 

Mitral Surgical Redo versus Transapical Transcatheter Mitral Valve Implantation 

Dear Dr. Zubarevich:

I'm pleased to inform you that your manuscript has been deemed suitable for publication in PLOS ONE. Congratulations! Your manuscript is now with our production department. 

Kind regards, 

on behalf of

Dr. Alessandro Parolari 

Academic Editor

PLOS ONE